# Assessing forest properties with data-driven vegetation indices: insights from 900,000 forest stands

Samuel M. Fischer<sup>1</sup>, Rico Fischer<sup>2</sup>, and Andreas Huth<sup>1,3,4</sup>

**Correspondence:** Samuel M. Fischer (samuel.fischer@ufz.de)

Abstract. Vegetation indices (VIs) are widely used to assess forest properties, but deriving VIs for attributes not mechanistically linked to forests' solar reflectance is challenging. Here, data-driven VIs could help, which yield information based on correlations identified in large datasets of forest and reflectance data. However, data-driven VIs are prone to bias and overfitting if data is limited and the functional form and wavelengths used for the VIs are not sensibly constrained. In this study, we facilitate the development of data-driven VIs by systematically analyzing VIs with two wavelengths (400 nm- 2400 nm) and evaluating their correlations to biomass, leaf area index (LAI), gross primary production (GPP), and net primary production (NPP) subject to different sources of environmental and physiological uncertainty. Considering 900, 000 forest stands simulated via a forest and radiative transfer modelling approach, we introduced a new class of VIs and found that data-driven VIs can provide highly accurate estimates. Particularly VIs combining near and shortwave infrared light yielded promising results, with biomass, LAI, and GPP often being well estimable from the same wavelength combinations; visible light gained importance in less dense and structurally heterogeneous forests. Both the functional form of the VIs and the considered uncertainty factors did not primarily reduce the achievable accuracy, but instead constrained the range of wavelengths from which good indices could be constructed. This suggests that data-driven vegetation indices can yield valuable results if the wavelength choice is optimized. This opens new pathways for utilizing recent hyperspectral satellite missions such as EnMAP.

## 5 1 Introduction

The reflectance spectrum of a forest contains a vast array of information about its structure, productivity, and health. With the increasing availability of large-scale datasets of remotely sensed multi-spectral images, vegetation indices have been developed to estimate key vegetation characteristics, such as biophysical, biochemical, and physiological plants properties (Zeng et al., 2022). These properties and the corresponding indices are often linked to forest stand characteristics such as leaf area index (LAI) or gross productivity (GPP), but the links can become inconclusive (e.g. in dense forests; Mutanga et al., 2023), or be weak for some forest characteristics of interest (e.g. net primary production; NPP; Xiao et al., 2019). Here, data from

<sup>&</sup>lt;sup>1</sup>Helmholtz Centre for Environmental Research – UFZ, Dept. of Ecological Modelling, Permoserstr. 15, 04318 Leipzig, Germany.

<sup>&</sup>lt;sup>2</sup>Julius Kühn-Institute (JKI) - Federal Research Center for Cultivated Plants, Institute for Forest Protection, Erwin-Baur-Str. 27, 06484 Quedlinburg, Germany.

<sup>&</sup>lt;sup>3</sup>Osnabrück University, Institute of Environmental Systems Research, Barbarastr. 12, 49076 Osnabrück, Germany.

<sup>&</sup>lt;sup>4</sup>German Centre for Integrative Biodiversity Research (iDiv) Halle-Jena-Leipzig, Puschstr. 4, 04103 Leipzig, Germany.

new hyperspectral satellite missions, such as the German EnMAP (Environmental Mapping and Analysis Program) satellite launched in 2022 (Guanter et al., 2015; Qian, 2021; Storch et al., 2023), could open new ways forward, as they contain information on not only a few but more than 250 different wavelength channels. Hyperspectral data could be used to develop new data-driven vegetation indices, harnessing subtle correlations between solar reflectance spectra and forest stand attributes, which may occur only in specific wavelength regions or require considering several wavelength channels jointly.

However, developing such vegetation indices is challenging, as forests' reflectance spectra are affected by heterogeneity in forest structure, plant traits, and environmental conditions (Dong et al., 2024). This variability may not only induce uncertainty when applying vegetation indices, but may also lead to overfitting when training the underlying models, unless the degrees of freedom for training are reduced by careful choice of the wavelengths to consider and the functional form to use for the index. These are the issues we address in this paper.

To that end, large-scale datasets covering the whole range of possible forests under potential environmental conditions and plant traits are required. These datasets, however, are often not available for NPP and other forest characteristics of interest. A common approach (see e.g. Fang, 2003; Lemaire et al., 2008; Wang et al., 2022; Dong et al., 2024) to bridge such data gaps is to consider reflectance data obtained from radiative transfer models (Jacquemoud et al., 2009; Ligot et al., 2014), which simulate canopy reflectance based on parameters capturing aspects of the environment, vegetation structure, and traits. This has the advantage that a wide range of scenarios can be considered easily, potentially leading to more general results than site-specific studies.

Since radiative transfer models for plant canopies include leaf attributes and the LAI as input parameters, several simulation studies have analyzed the relationships between these variables and reflectance spectra (Fang, 2003; Lemaire et al., 2008; Estévez et al., 2020). An analysis considering stand characteristics such as NPP or structural properties, in turn, requires a model representation of forest dynamics as well. We address this problem via a novel hybrid simulation approach connecting an individual-based forest model with a radiative transfer model (Henniger et al., 2023a).

The history of individual-based forest models dates back to the 70s of the past century (Botkin et al., 1972) and has led to the development of a wide spectrum of models varying in application focus and complexity (Bugmann and Seidl, 2022). Though radiative transfer models for canopy reflectance have a similarly long history (Allen et al., 1970; Verhoef, 1984), the two model types have rarely been combined until recently (Henniger et al., 2023a). In this study, we use the process-based forest gap model FORMIND (Bohn et al., 2014; Fischer et al., 2016), which considers trees on the individual level and computes their gross and net productivity based on environmental conditions (e.g. temperature and precipitation), interactions between trees (e.g. light and water competition), as well as species-specific traits. As the model represents productivity and carbon fluxes explicitly and features both individual trees and their vertical leaf distribution, FORMIND is particularly suited to assess forest productivity and structure (Bohn and Huth, 2017; Fischer et al., 2024).

FORMIND can be tightly coupled with the multi-layer radiative transfer model mSCOPE (Yang et al., 2017). The mSCOPE model simulates four streams of sunlight (direct and diffuse down- and upstream of light) in the wavelength range 400 nm to 2500 nm and extends the classical SCOPE ("Soil Canopy Observation, Photochemistry and Energy fluxes") model (van der Tol et al., 2009) by facilitating the simulation of reflectance spectra of canopies with vertically heterogeneous leaf traits. The

https://doi.org/10.5194/egusphere-2025-5198 Preprint. Discussion started: 10 November 2025

© Author(s) 2025. CC BY 4.0 License.

coupled model version allows analyzing the reflectance spectra of dynamically evolving forests and includes an extension of mSCOPE for multi-species forests (Henniger et al., 2023a). Here, we combine it with a Monte-Carlo approach – an updated version of the "Forest Factory" (Henniger et al., 2023b) – to generate and analyze a total of 900,000 central European forest stands along with their reflectance profiles subject to different sources of uncertainty and noise.

The dataset we obtained (and publish along with this study) contains reflectance data for more than 250 different wavelength channels and corresponding forest properties. Based on these data, we derive specialized vegetation indices for estimating the forest characteristics biomass, LAI, GPP, and NPP. We conduct a systematic analysis of all vegetation indices combining two wavelengths in the 400 nm to 2400 nm range, evaluating their ability to estimate biomass, LAI, GPP, and NPP and comparing the quality of the obtained estimates to predictions derived from "classical" vegetation indices. As vegetation indices come in countless varieties of functional forms – each potentially constraining the accuracy – we introduce a new class of vegetation indices, called "parameter-free" vegetation indices, whose form is not pre-imposed via a parametric function but rather directly inferred from the data. We evaluate which wavelength combinations are best suited for estimating each of the considered forest characteristics and which form the optimal models take on. Furthermore, we analyze how these results are affected by (1) the presence of different sources of environmental and physiological uncertainty or (2) the structure of the considered forests.

This paper is structured as follows: first, we provide an overview of our approach. Then we show how we parameterized the forest model and generated the analyzed forest stands. Afterwards, we describe the parameterization of the radiative transfer model and adjustments we introduced to simulate the reflectance profiles of forests with a vertically heterogeneous leaf density. Finally, we provide details on the analysis of the generated forest stands and present the results.

#### 75 2 Methods

We analyzed the relationships between forests' solar reflectance spectra and their biomass, LAI, GPP, and NPP in different uncertainty scenarios. For each scenario, we generated 100,000 forest states via a Markow Chain Monte Carlo (MCMC) approach applied to a forest model, and simulated the corresponding reflectance spectra using the radiative transfer model mSCOPE (Yang et al., 2017). Then we fitted parameter-free models to investigate the links between the forest characteristics and the reflectance values at different wavelengths and compared their predictive capabilities to those of existing vegetation indices. To investigate how different sources of uncertainty affect the investigated relationships, we applied different noise terms to the parameters of the radiative transfer model in each scenario and analyzed their effects on the results. The process is depicted as a flow chart in Fig. 1. Below we provide details on each of these steps.

### 2.1 Generating forest stands

We used the process-based forest model FORMIND along with a MCMC approach to generate a sample of forest stands representing the variety of forest compositions potentially found in the field, building on the approach by Henniger et al. (2023b). FORMIND estimates the GPP and respiration of individual trees by considering competition for light and water and incorporating environmental factors such as temperature and precipitation. Before applying the model, we updated its parameterization

**Figure 1.** Flow chart depicting the development of parameter-free vegetation indices for biomass, leaf area index (LAI), gross primary production (GPP), and net primary production (NPP). The entropy of tree diameters (DBH entropy) is computed as measure of structural diversity to analyze the relationship between forest structure and optimal vegetation indices.

for temperate forests in central Europe to better capture the carbon dynamics of heterogeneous forests. We considered forest inventory data collected at the "Hohes Holz" research site in central northern Germany (mean ann. temperature  $10.8\,^{\circ}$ C; mean ann. precipitation  $6601/\text{m}^2$ ) and changed the temperate model parameterization (Bohn et al., 2014) in the following aspects: (1) we updated the allometric relationships according to data from Jucker et al. (2022); (2) we adjusted parameters for light climate and respiratory losses to better model the carbon dynamics in structurally heterogeneous forests; (3) we chose parameters on soil properties that matched the soil at the research site (values taken from Maidment, 1993); (4) we added the mechanistic defoliation and mortality mechanism introduced by Fischer et al. (2024); and (5) we used a new approach to consider weather data on a daily time scale whereas forest dynamics such as growth and mortality remain modelled on the yearly time scale. Details on the parameterization can be found in SI S5.

Typically, forest models such as FORMIND are initialized with existing forest inventory data or simulate forest successions starting from bare ground. However, since these simulations may not cover the entirety of forest states found in managed forests, we used a different approach here. Trees cannot survive if their GPP is insufficient to cover their respiratory needs.

110

Hence, assuming that the GPP of any live tree found in a real forest must exceed its respiration (Bohn and Huth, 2017), we randomly assembled a sample of states satisfying this property, starting from a uniform prior of forest states.

We considered forest patches of  $20 \,\mathrm{m} \times 20 \,\mathrm{m}$  size. Starting from bare ground, we randomly added or removed trees of random sizes and species. Then we computed the GPP and autotrophic respiration of the forest stand for one year and evaluated whether the stand was "feasible", i.e., (1) each tree's GPP exceeded its respiration, and (2) the tree crowns did not exceed the available space. If the stand was feasible, we repeated the procedure; otherwise we returned to the last feasible state and repeated the attempt to add and remove trees until a feasible state was found.

To increase the range of qualitatively different forest stands, we used a hierarchical sampling approach when adding new trees, randomly constraining the potential tree sizes and species in some stands (Henniger et al., 2023b). Furthermore, we facilitated computational efficiency by dynamically adjusting the "step size", i.e., the number of individual tree additions or removals between two forest feasibility checks: we started at 5 additions / removals and increased or decreased this number by factor 2 if the last five feasibility checks were positive or negative, respectively. After 200 tree additions or removals, we terminated the process. Further details regarding this approach are provided in SI S2.

We generated 100,000 forest stands for the environmental conditions found at the Hohes Holz site. The forest stands differ, among others, in biomass, tree number, and species composition. For each stand, we computed the biomass, LAI, GPP, and NPP (mean values:  $321 \frac{\text{tODM}}{\text{ha}}$  biomass, LAI of 3,  $28.8 \frac{\text{tODM}}{\text{ha} \cdot \text{yr}}$  GPP,  $3.4 \frac{\text{tODM}}{\text{ha} \cdot \text{yr}}$  NPP). Furthermore, we determined the basal-area-weighted DBH entropy for the tree size distribution (Fischer et al., 2024), which is a measure for the structural heterogeneity of forests (mean: -4.54). We provide the formula of basal-area-weighted DBH entropy in SI S4. The distribution of these forest characteristics is visualized in SI S3. Lastly, we computed the forests' solar reflectance spectra using the radiative transfer model mSCOPE, as described below.

#### 2.2 Simulating solar reflectance spectra

We simulated the solar reflectance spectra of the generated forest patches for wavelengths between 400 nm and 2400 nm. To that end, we used an extended version of the radiative transfer model mSCOPE, designed for application in forests with vertically heterogeneous leaf traits (Yang et al., 2017; Henniger et al., 2023a). Our extended model version adds on to the original in also considering vertically heterogeneous leaf *densities*. Furthermore, we adjusted the probability of observing sunflecks on the ground, as the structure and covering of the ground are difficult to model exactly.

We parameterized the radiative transfer model similar to Henniger et al. (2023a) but adjusted individual parameters based on data from the TRY plant trait database (Kattge et al., 2011). A list of the parameters can be found in SI S1. We set the view zenith angle to  $0^{\circ}$  and the sun zenith angle to  $31.5^{\circ}$ , corresponding to maximal zenith angle observed at the Hohes Holz site. Furthermore, we set the relative azimuth angle between sun and view direction to a value of  $140^{\circ}$  (Henniger et al., 2023a), but note that this value is only relevant if the view zenith is changed from its default  $0^{\circ}$ . We set the probability to observe direct sunlight (sunflecks) at the ground to 0, which reduced the simulated reflectance values in the visible range. This improved the agreement between model results and field observations at the study site.

We simulated forests and corresponding hyperspectral data for 9 scenarios with different sources of of uncertainty each. First, we considered the scenario without any parameter uncertainty. Second, we adjusted the LAI input to the radiative transfer model in each simulated forest patch by a random factor  $\varepsilon \in \left[\frac{2}{3}, \frac{3}{2}\right]$  from a log-uniform distribution:  $\ln(\varepsilon) \sim \mathcal{U}(\ln(2/3), \ln(3/2))$ . Here,  $\mathcal{U}(a,b)$  is the uniform distribution in [a,b]. Using a log-uniform multiplicative perturbation ensures that the parameters remain in the positive domain and in their respective order of magnitude.

Third, we perturbed all species' leaf trait parameters (Chlorophyll a+b content, leaf mass per unit area, equivalent water thickness, senescence material, carotenoid content) independently by random factors  $\varepsilon \in \left[\frac{2}{3}, \frac{3}{2}\right]$  (log-uniform distribution). Fourth, we altered the leaf structure parameter in each patch by a random factor  $\varepsilon \in \left[\frac{2}{3}, \frac{3}{2}\right]$  (log-uniform distribution), but constrained the resulting values to the admissible range [1,3] where necessary. Fifth, we changed the leaf inclination parameters by random values  $\epsilon \sim \mathcal{U}(-0.2, 0.2)$  and constrained them to the admissible range [-1,1] if required. Furthermore, we normalized these parameters by their joint Euclidean norm if the norm exceeded 1.

Sixth, we drew the soil wetness parameter in each simulated patch randomly from  $\mathcal{U}(0,1)$ . Seventh, we changed the sun and view zenith angles as well as the relative azimuth by independent random values from  $\mathcal{U}(-15^{\circ}, 15^{\circ})$ . Eighth, we imposed independent random perturbations to all simulated reflectance values. The perturbations followed normal distributions with mean 0 and standard  $\sigma_i = 0.05R_i + 0.005$ , where i indicates the wavelength and  $R_i$  the corresponding reflectance value. Finally, ninth, we combined all the perturbations listed above.

## 2.3 Analyzing vegetation indices

For each considered uncertainty scenario, we generated 100,000 forest states with corresponding reflectance spectra and filtered out all forest states with a biomass below  $50 \frac{\text{t} \, \text{ODM}}{\text{ha}}$ , as the hyperspectral model was primarily designed for areas with full forest cover. We split the datasets into 75% training and 25% validation data. Then we selected all possible pairs of two wavelengths and fitted a model estimating either biomass, LAI, GPP, or NPP based on the corresponding reflectance values of these wavelengths in the training data.

Here, we considered two types of models: (1) an affine-linear model of the form  $f = a_0 + a_1 R_{w_1} + a_2 R_{w_2}$ , where f is the forest attribute,  $w_1$  and  $w_2$  are the wavelengths, and  $R_{w_1}$  and  $R_{w_2}$  are the corresponding reflectance values, and (2) a parameter-free model based on regular binning of the data. The shape of parameter-free models is not given by a simple formula but directly derived from the data, making these models particularly useful for data-driven vegetation indices. To obtain the parameter-free model, we binned the data into 25 regular intervals along each wavelength axis, resulting in 625 potential bins for the tuples  $(R_{w_1}, R_{w_2})$ . Then we determined the mean vale of the considered forest property in each bin and assigned this value to the middle point of this bin. That is, for a bin  $[a_{w_1}, b_{w_1}] \times [a_{w_2}, b_{w_2}]$ , we set

$$f\left(\frac{a_{w_1} + b_{w_1}}{2}, \frac{a_{w_2} + b_{w_2}}{2}\right) := \frac{1}{|\mathcal{B}|} \sum_{i \in \mathcal{B}} f_i,\tag{1}$$

where  $\mathcal{B}$  is the set of all data points in the bin and  $|\mathcal{B}|$  its cardinality. We then applied a bi-linear interpolation between these points to obtain model predictions for all other reflectance values. For reflectances outside the convex hull of the training data, we applied a nearest neighbour extrapolation based on the closest bin centre.

| Range name                   | Wavelength range [nm] | Wavelength used in analysis of classical indices [nm] |  |  |
|------------------------------|-----------------------|-------------------------------------------------------|--|--|
| Ultraviolet (UV)             | [400, 450)            | -                                                     |  |  |
| Blue                         | [450, 500)            | 460                                                   |  |  |
| Green                        | [500, 570)            | 560                                                   |  |  |
| Red                          | [570, 670)            | 655                                                   |  |  |
| Red edge (RE)                | [670, 760)            | 725                                                   |  |  |
| Near infrared (NIR)          | [760, 1360]           | 865                                                   |  |  |
| Shortwave infrared 1 (SWIR1) | [1440, 1820]          | _                                                     |  |  |
| Shortwave infrared 2 (SWIR2) | [1950, 2400]          | _                                                     |  |  |

**Table 1.** Considered wavelength ranges. The water absorption bands are excluded, as no reliable reflectance data can be obtained for these wavelengths in practice. The last column provides the wavelengths used for representing the band when analyzing classical vegetation indices (Tab. 2).

For both model types, we computed the Pearson  $R^2$  of each fitted model based on the validation data. Furthermore, we created heatmaps for the  $R^2$  values as function of the wavelengths. Afterwards, we determined for each forest characteristic the maximal obtained  $R^2$  and the percentage  $P_{0.9}$  of models achieving an  $R^2$  of at least 90% of the maximum. We computed  $P_{0.9}$  by determining the surface area in the heatmap where models achieved a sufficiently large  $R^2$  and dividing it by the total surface area of all considered bands (Tab. 1). Specifically, we defined a function  $f_{0.9}(w_1, w_2)$  with  $f_{0.9}(w_1, w_2) = 1$  for wavelength pairs with sufficient  $R^2$  and  $f_{0.9}(w_1, w_2) = 0$  otherwise, Then,  $P_{0.9} = \int \int f_{0.9}(w_1, w_2) dw_1 dw_2$ , which we approximated via a midpoint Riemann sum (numerical integration) of  $f_{0.9}$  with those wavelength pairs as sampling points for which we had computed reflectance values.

To measure how important each individual band (Tab. 1) is for obtaining a large  $R^2$ , we determined the fraction of all models that use a wavelength from the respective band and achieve a high  $R^2$  (90% of max). We plotted this quantity (colour coded by band) along with the respective maximal  $R^2$  for each considered forest characteristic and uncertainty scenario.

To understand the impact of forest structure on these results, we repeated the analysis in the scenario without uncertainty after grouping the data by forest density (biomass below or greater than  $200 \frac{\text{t} \, \text{ODM}}{\text{ha}}$ ) and structural diversity (DBH entropy  $S_{\text{DBH}}$  less or greater than -2.5; cf. Fischer et al., 2024). That way, we obtained the maximal  $R^2$  values and the most significant wavelength bands for forests with different structural properties. To streamline our study, we focused our analysis on the NPP.

We visualized the functional form of the best models in the absence of noise with heatmaps. We set the obtained  $R^2$  values into context with existing vegetation indices by determining the correlations of classical vegetation indices with biomass, LAI, GPP, and NPP based on our simulated data without noise. Here, we considered all vegetation indices listed by Zeng et al. (2022) that use reflectances from clearly defined wavelengths or bands and do not depend on a site-specific parameterization. By squaring the correlation coefficients, we obtained the  $R^2$  values that could potentially be achieved with affine-linear models predicting the forest properties based on the vegetation indices.

#### 3 Results

185

The  $R^2$  values of the analyzed classical vegetation indices and the best linear and parameter-free models are displayed in Tab. 2. For the scenario without uncertainty, the new data-driven models achieve much higher  $R^2$  values than the best classical vegetation indices: 0.75 (data-driven) vs. 0.34 (NDVIre) for the biomass, 0.97 (data-driven) vs. 0.6 (EVI) for the LAI, 0.75 (data-driven) vs. 0.55 (EVI) for the GPP, and 0.6 (data-driven) vs. 0.34 (CAI) for the NPP. The maximal  $R^2$  achieved with linear models was almost equal to the value achieved with parameter-free models and in one case (GPP) even slightly higher.

While the maximal achieved  $R^2$  values were similar for linear and parameter-free models, the range  $P_{0.9}$  of wavelengths for which high  $R^2$  values (90% of max.) were achieved differed and was much smaller for linear models, especially for biomass and NPP (Fig. 2). In the scenario without environmental or physiological uncertainty, the biomass and LAI could be estimated best based on NIR reflectance combined with a reflectance from narrow bands from the SWIR range: one band close to the water absorption band between NIR and SWIR1, one at the centre of SWIR1, one close to the water absorption band between SWIR1 and SWIR2, and one encompassing the second half of SWIR2. These wavelength ranges were also well suited for estimating GPP, for which, however, an additional range combining green and red with NIR wavelengths yielded high  $R^2$  values as well.

NPP was best estimable based on wavelengths in the SWIR range combined with any other range. Specifically, combinations of the centre SWIR1 range with any visible or NIR wavelength, combinations of two significantly different SWIR1 wavelengths, and combinations of SWIR2 with visible wavelengths or SWIR1 wavelengths close to the first water absorption band permitted high  $R^2$  values.

Analyzing the best models for the considered forest characteristics yielded several similarities. The best models have in common that the estimated forest properties increase with the reflectance of one wavelength, while they decrease with the reflectance of the other (Fig. 3). The reflectance values we used for fitting the models were correlated and hence accumulated along increasing lines  $l: R_{w_2} = a_0 + a_2 R_{w_1}$  in the two-wavelength space (see dots in Fig. 3;  $R_{w_1}$  and  $R_{w_2}$  are the reflectances of the considered wavelengths  $w_1$  and  $w_2$ ). The gradient of the forest properties was perpendicular to this line, indicating that (1) many different reflectance pairs could lead to the same forest property estimate if they lie on a line parallel to the correlation line l and (2) the models were sensitive to changes in individual reflectance values (other reflectance held constant).

The effect of uncertainty on the potential  $R^2$  was moderate in most of the considered scenarios (dots in Fig. 4). In five of the seven considered noise scenarios with only one uncertainty source, the maximal obtained  $R^2$  was almost insensitive to noise

| Index             | Full name                                                  | Equation                                                                                                                    | $R^2$   |       |       |       |
|-------------------|------------------------------------------------------------|-----------------------------------------------------------------------------------------------------------------------------|---------|-------|-------|-------|
| Index             | Turi name                                                  | Equation                                                                                                                    | Biomass | LAI   | GPP   | NPP   |
| SR                | Simple ratio                                               | NIR/Red                                                                                                                     | 0.28    | 0.28  | 0.34  | 0.01  |
| NDVI              | Normalized difference vegetation index                     | $rac{	ext{NIR}-	ext{Red}}{	ext{NIR}+	ext{Red}}$                                                                            | 0.29    | 0.31  | 0.36  | 0.00  |
| MSR               | Modified simple ratio                                      | $\frac{\text{NIR/Red} - 1}{\sqrt{\text{NIR/Red} + 1}}$                                                                      | 0.29    | 0.29  | 0.35  | 0.01  |
| DVI               | Difference vegetation index                                | NIR - Red                                                                                                                   | 0.10    | 0.55  | 0.49  | 0.16  |
| EVI2              | Two-band enhanced vegetation index (EVI) without blue band | $\frac{2.5(\mathrm{NIR-Red})}{\mathrm{NIR}+2.4\cdot\mathrm{Red}+1}$                                                         | 0.12    | 0.58  | 0.53  | 0.15  |
| NIRv              | Near-infrared reflectance of vegetation                    | NDVI · NIR                                                                                                                  | 0.11    | 0.57  | 0.52  | 0.16  |
| kNDVI             | Kernel-normalized difference vegetation index              | $\tanh \left( \mathrm{NDVI^2} \right)$                                                                                      | 0.29    | 0.31  | 0.36  | 0.00  |
| EVI               | Enhanced vegetation index                                  | $\frac{2.5(\mathrm{NIR}\mathrm{-Red})}{\mathrm{NIR}\mathrm{+}6\cdot\mathrm{Red}\mathrm{-}7.5\cdot\mathrm{Blue}\mathrm{+}1}$ | 0.14    | 0.60* | 0.55* | 0.15  |
| PRI               | Photochemical reflectance index                            | $\frac{R_{531} - R_{570}}{R_{531} + R_{570}}$                                                                               | 0.10    | 0.01  | 0.00  | 0.07  |
| CCI               | Chlorophyll/carotenoid index                               | $\frac{R_{530} - R_{650}}{R_{530} + R_{650}}$                                                                               | 0.00    | 0.11  | 0.01  | 0.12  |
| GCC               | Green chromatic coordinate                                 | $\frac{\text{Green}}{\text{Red} + \text{Green} + \text{Blue}}$                                                              | 0.05    | 0.06  | 0.01  | 0.18  |
| CIred-edge        | Red-edge chlorophyll index                                 | NIR/RE-1                                                                                                                    | 0.33    | 0.28  | 0.44  | 0.00  |
| NDVIre            | Red-edge NDVI                                              | $rac{	ext{NIR} - 	ext{RE}}{	ext{NIR} + 	ext{RE}}$                                                                          | 0.34*   | 0.29  | 0.45  | 0.00  |
| MTCI              | MERIS total chlorophyll index                              | $\frac{R_{750} - R_{710}}{R_{710} - R_{680}}$                                                                               | 0.33    | 0.18  | 0.33  | 0.01  |
| NDWI <sup>†</sup> | Normalized difference water index                          | $\frac{R_{860} - R_{1240}}{R_{860} - R_{1240}}$                                                                             | 0.07    | 0.46  | 0.40  | 0.11  |
| NDLI              | Normalized difference lignin index                         | $\frac{\ln R_{1754} - \ln R_{1680}}{\ln R_{1754} + \ln R_{1680}}$                                                           | 0.13    | 0.58  | 0.51  | 0.12  |
| CAI               | Cellulose absorption index                                 | $100\left(\frac{R_{2019} + R_{2206}}{2} - R_{2109}\right)$                                                                  | 0.31    | 0.05  | 0.02  | 0.34* |
| _                 | Linear models                                              | $a_0 + a_1 R_{w_1} + a_2 R_{w_2}$                                                                                           | 0.74    | 0.96  | 0.75  | 0.59  |
|                   | Non-parameteric models                                     | n.a.                                                                                                                        | 0.75    | 0.97  | 0.74  | 0.60  |

Table 2. Squared correlation coefficients between different forest characteristics and vegetation indices listed in Zeng et al. (2022). The asterisks highlight the respective highest values in each column, excluding the values obtained by models developed in this paper, for which the best  $R^2$  values are given in the last two rows. The values were computed based on forests generated without considering environmental or physiological uncertainty. The formulas and index names were taken from Zeng et al. (2022) with minor corrections. The specific wavelengths used for the named bands are provided in Tab. 1.  $^{\dagger}$  Note on NDWI: version by Gao (1996), not McFeeters (1996).

Figure 2. Wavelength combinations used in the best (a) linear and (b) parameter-free models for different properties of forests without environmental or physiological uncertainty. Each colour corresponds to a different forest characteristic. The shaded areas show the combinations of wavelengths for which models with at least 90% the  $R^2$  value of the best 2-wavelength model could be constructed. The graphs on the x and y axis show the mean reflectance profiles of all considered forest stands.

and did not decrease by more than 0.07. For biomass, LAI, and GPP, this applied for uncertainty in leaf traits, leaf structure, leaf angle, soil wetness, and sun and view angles. White noise had a slightly higher effect ( $R^2$  reduction by up to 0.15), and LAI uncertainty was the individual uncertainty source with the highest impact ( $R^2$  reduction up to 0.2 for biomass and GPP and 0.3 for LAI). For the NPP, LAI uncertainty was less significant for the maximal  $R^2$ . Instead, the leaf parameters played the most important individual role ( $R^2$  decrease by 0.16). For all forest characteristics, the combined noise scenario yielded the lowest  $R^2$ , which was about half as big as without uncertainty, respectively.

The ranges of wavelengths that yield a near-optimal  $R^2$  (bars in Fig. 4) were more strongly affected by uncertainty than the optimal potential  $R^2$  values. Here, uncertainty in the leaf traits and the leaf structure had the strongest effect, in particular for biomass and NPP, where the fraction  $P_{0.9}$  of evaluated models with an  $R^2$  exceeding 90% of the optimum dropped by more than 82%. For the NPP, uncertainty in the leaf inclination parameters had a similarly strong impact. The value  $P_{0.9}$  decreased least in the presence of LAI and sun / view angle uncertainty. In the joint noise scenario,  $P_{0.9}$  was strongly reduced for all forest characteristics.

In most considered scenarios, the presence of uncertainty did not affect the combinations of wavelength bands in which models with a high  $R^2$  could be found (colours in Fig. 4). However, in the presence of leaf trait uncertainty, no models using

Figure 3. The best parameter-free models for (a) biomass, (b) LAI, (c) GPP, and (d) NPP, respectively, for forests without environmental or physiological uncertainty. The axes correspond to reflectance values ( $R_w$  is the reflectance of wavelength w in nanometers). The background colour depicts the model's prediction, i.e., the respective forest characteristic corresponding to a reflectance pair. The points correspond to simulated forest patches in the validation dataset: their positions correspond to their reflectances, their colours to their biomass, LAI, GPP, and NPP.

visible and red edge bands achieved near-optimal  $\mathbb{R}^2$  values. For the NPP, uncertainty in the leaf structure parameters had a similar effect. Furthermore, models with near-infrared light lost their predictive capabilities for NPP if uncertainty beyond LAI and sun / view angles was present.

Figure 4. Impact of different uncertainty factors on the predictability of forest characteristics. Each bar and point corresponds to a source of uncertainty. The circles depict the maximal possible  $R^2$  values. The bars show the fraction of considered wavelength pairs for which a model with an  $R^2$  of at least 90% of the respective maximum can be constructed. Here, the colours depict the bands from which the corresponding wavelengths are taken. It is visible that while the maximal  $R^2$  is only moderately sensitive to uncertainty in environmental factors and leaf properties, the range of wavelengths for which a high  $R^2$  can be attained decreases significantly in the presence of noise.

Filtering the considered forest stands and developing specialized models for forests with high/low biomass and/or structural diversity generally improved the  $R^2$  values for NPP (black circles in Fig. 5). NPP estimates achieved particularly high  $R^2$  values (> 0.7) in forests with small biomass (

forests with low biomass and low structural heterogeneity, whereas the  $\mathbb{R}^2$  dropped to 0.49 in forests structurally diverse forests with high biomass. If no biomass filter was applied, however, NPP could be better estimated in structurally diverse forests.

Structural diversity had a larger effect than the biomass on  $P_{0.9}$ , the range of wavelengths based on which NPP estimates with a high  $R^2$  ( $\geq 90\%$  of max.) could be achieved (coloured circles in Fig. 5). More models achieved a near-maximal  $R^2$  in structurally homogeneous forests. The biomass range only made a strong difference for structurally homogeneous forests, where  $P_{0.9}$  was significantly larger in forests with a low biomass.

Filtering by biomass and structural diversity had a significant effect on the bands that were best suited to estimate NPP (colours in Fig. 5). Most prominently, the percentage of high-achieving models using NIR light was much larger in forests known to have a small biomass (namely 51%-70%) or large structural heterogeneity (20%-99%). In contrast, NIR light was much less used in the best models for unfiltered forests (14% of the high-achieving models) or forests with neither large structural heterogeneity nor small biomass (0%).

#### 4 Discussion

We analyzed 900,000 temperate forest stands and corresponding reflectance profiles. Considering different sources of uncertainty, we systematically evaluated a total of more than 7 million data-driven models for biomass, LAI, GPP, and NPP based on different wavelength pairs. We found that even in the presence of significant uncertainty, the data-driven models yielded relatively large  $R^2$  values, significantly exceeding the  $R^2$  values obtained from classical vegetation indices. This suggests that data-driven vegetation indices, even if using only two wavelengths, are a promising tool for deriving forest properties from remote sensing data.

Neither a large number of different wavelengths nor a complicated functional form were necessary to estimate forest attributes with relatively high accuracy. In fact, linear models superseded parameter-free models without pre-imposed functional form in some wavelength regions. However, focusing on a specific functional form limited the range of wavelengths suitable for analyzing forest attributes and increased the challenge of identifying the optimal wavelengths. This agrees with earlier findings by Gong et al. (2003), who evaluated different wavelength combinations and functional forms to estimate LAI based on reflectance data. The better suitability of linear models in some wavelength regions is due to the limited resolution of the binning method we applied in the parameter-free case, where we computed average values of forest characteristics for individual reflectance intervals. Increasing the resolution (i.e., using smaller intervals) or using more general parametric models will therefore lead to even better  $\mathbb{R}^2$  values than we presented, but may require larger datasets.

Despite the relatively high  $R^2$  values we obtained, ranging from 0.6 for the NPP to 0.97 for the LAI, our results show that identifying the right wavelengths for estimating specific forest properties is key. In the absence of uncertainty, the classical vegetation indices rarely utilized wavelength pairs optimal for estimating any of the considered forest characteristics. Exceptions were the vegetation indices using red and NIR / red edge light (e.g. NDVI, EVI, NDVIre, NIRv), which were, however, only in the optimal range for estimating GPP and neither of the other forest characteristics. This is surprising, since LAI and

Figure 5. Predictability of the NPP for subsets of the forest stands. In the second and third column, the forest stands are filtered by DBH entropy, where low values indicate dominance by a single tree or multiple similarly-sized trees, whereas high values indicate that the basal area is evenly distributed over trees of different size classes. In the first column, no filtering by DBH entropy is applied. Similarly, the second and third row correspond to forests with high and low above-ground biomass, respectively, whereas no filtering was applied in the first row. The diameters of the hollow black circles correspond to the respective maximal  $R^2$  values that could be achieved for the dataset. The shaded areas are proportional to the fraction of considered wavelength pairs for which a high  $R^2$  (> 90% of maximum) could be achieved. The colours depict the bands from which the corresponding wavelengths were taken. It is visible that for dense forests, most models consider a wavelength from the near infrared band. In contrast, for sparser forests, the SWIR1 band was used more frequently. In general, it was easier to estimate the NPP of dense forests and forests dominated by trees from few size classes.

GPP are typically assumed to be strongly correlated (Gitelson et al., 2014), and several vegetation indices were developed by considering the reflectance profile of leafs (Zeng et al., 2022).

In the presence of environmental or physiological uncertainty, the range of suitable wavelengths decreased significantly, and some bands lost their suitability. For example, visible light became sub-optimal when leaf traits were uncertain, and NIR light became unsuited for estimating NPP in this case. This suggests that the choice of wavelengths for estimating forest characteristics should take into account the type and magnitude of uncertainty expected in the field data. Identifying and

quantifying these sources of uncertainty will hence be an important preparational step to deriving data-driven vegetation indices for forest characteristics.

Aside the above-mentioned cases, the best band combinations for estimating forest characteristics remained remarkably stable throughout the considered scenarios. In particular, combinations of longer wavelengths (NIR, SWIR1, and SWIR2) were generally well suited. This is in line with previous studies (Gong et al., 2003; Lemaire et al., 2008; Psomas et al., 2011; Houborg and McCabe, 2018; Almeida et al., 2019), yet contrasts with the most common choices of wavelengths for vegetation indices (only 4 out of 23 vegetation indices listed in the review of Zeng et al. (2022) combined two wavelengths from these bands). However, these vegetation indices may also applied to assess the *presence* of vegetation, e.g. classify areas into forest landscapes and other landscape types, whereas in our study, we considered forests only, i.e., presumed that the landscape had been filtered to only include forests before. Nonetheless, a classification into forested and unforested land could be easily conducted in an independent preparational step, making forest-specific vegetation indices applicable in combined multi-step procedures in practice.

We found that combinations of NIR and SWIR wavelengths were particularly useful for estimating forest characteristics. Reflectance in the SWIR range is strongly related to leaf water content, lignin, proteins, nitrogen and cellulose (Curran, 1989; Fu et al., 2021; Zeng et al., 2022), and the combination of LAI and nitrogen content has been found well suited for estimating forest productivity (Reich, 2012; Zhang et al., 2023). The significance of NIR / SWIR combinations may be further understood by considering the sensitivity of forest reflectance to LAI, which is low in the NIR range but high in the SWIR range (Verrelst et al., 2015; sensitivity analysis of radiative transfer models). Furthermore, NIR and SWIR reflectances are much less sensitive to leaf constituents such as chlorophyll or carotenoid than reflectances from the visible spectrum (Mousivand et al., 2014; Verrelst et al., 2015; Prikaziuk and van der Tol, 2019). This may also explain why visible light became unsuitable for estimating GPP when the leaf properties were uncertain. We note, however, that a discussion of potential technical limitations in measuring NIR / SWIR light or required atmospheric corrections is beyond the scope of this study.

Since uncertainty of leaf constituents may be induced by high species richness, reflectances from the visible spectrum may be unsuitable for assessing particularly diverse forests. The significance of visible light decreased similarly in forests with high biomass, potentially due to the low penetration depth of visible light (Hovi and Rautiainen, 2020) leading to a quick saturation of reflectance-LAI relationships (Mutanga et al., 2023). These findings suggest that visible light is sub-optimal to assess the considered forest characteristics in structurally complex forests.

Noteworthily, an increase in the heterogeneity of tree sizes had a contrasting effect: it increased the significance of visible light for estimating NPP. The tree size diversity, measured by the DBH entropy in this study, is low if forest patches are dominated by individual large trees, overshadowing understorey trees. If multiple differently sized trees contribute equally to a forest's basal area, it is more likely that their canopies are directly visible from above, making it easier and potentially optimal to use visible light to assess the forest state. This relationship may not hold in strongly stratified (e.g. tropical) forests, though.

Our model-based approach allowed us to investigate the relationships between forests' solar reflectance spectra and forest structure and productivity for a large number of different forest stands. Nonetheless, this approach also came with drawbacks due to inherent model limitations. For example, though considering environmental effects such as drought, the impact of

leaf phenology or trees' health and stress level on optical leaf properties (Watt et al., 2021; Zhou et al., 2021) are currently not captured by the forest model used here (FORMIND). Similarly, the effects of shadows occurring on the surface of heterogeneous tree canopies (Hilker et al., 2010; Zeng et al., 2022), which could affect the estimability of forest characteristics by inducing a relationship between forests' height heterogeneity and their reflectance, are not accounted for in mSCOPE. We addressed these simplifications by modelling and investigating the effect of different uncertainty factors on the results, and our findings are in good agreement with earlier analyses on the predictive power of wavelength combinations for estimating LAI (Gong et al., 2003).

Building on previous work on generating ecologically feasible forests with simulation models (Bohn and Huth, 2017; Henniger et al., 2023b) and combining forest models with radiative transfer models (Henniger et al., 2023a), we introduced several methodological advancements. Our Markov Chain Monte Carlo approach for sampling forests allows a rigorous statistical interpretation of the distribution of the resulting forests. Furthermore, the algorithm's simplicity reduces the risk of model artifacts potentially favouring specific forest structures, makes it easier to incorporate climatic effects on forest states and to exchange the underlying forest model with other process-based forest models. The updates of the forest model FORMIND and its parameterization for temperate forests improved the model's efficiency and accuracy in heterogeneous forests.

The adjusted version of mSCOPE disentangles the vertical space and LAI dimensions, thereby permitting the simulation of forests with heterogeneous leaf densities in different layers of the canopy. The new soil sunfleck parameter reduces the impact of uncertain soil properties on reflectance and may lead to more accurate results in scenarios with large solar zenith angles.

Combining a process-based forest model with a radiative transfer model permits new lines of research on the relationships between forest attributes and forests' solar reflectance profiles. In this study, we focused on the forest attributes biomass, LAI, GPP, and NPP, which are key for understanding forest dynamics. However, our approach can easily be extended to conduct similar analyses for other forest characteristics such as species and structural diversity, net ecosystem exchange, forest health or disturbance. The results, such as the  $R^2$  maps presented in this study, could inspire new data-driven vegetation indices specialized for specific forest characteristics.

Moreover, since the composition of forests and their attributes can be configured freely, tailor-made vegetation indices could be designed for forests with already known structural characteristics. In this study, we aimed at considering a broad spectrum of forests with different structural characteristics. Via the random pre-sampling of species pool and height range (Henniger et al., 2023b), we obtained even-aged monocultures as well as uneven-aged heterogeneous forests. If the area of interest contained forests with known characteristics, such as some highly managed forests in central Europe, tailor-made vegetation indices could yield even better results. The data set of combined forest and reflectance data that is published along with this study can be easily utilized for corresponding analyses.

In this study, we focused on simple models considering only two wavelengths each. This facilitated a deeper understanding of the importance of individual wavelengths for assessing forests. Applied to hyperspectral data from the field, the presented approach could lead to the development of new vegetation indices. These could be used to assemble novel large-scale datasets on forest attributes, increasing our understanding of the state and dynamics of forests on the global scale.

https://doi.org/10.5194/egusphere-2025-5198 Preprint. Discussion started: 10 November 2025

© Author(s) 2025. CC BY 4.0 License.

#### 5 Conclusions

We used a hybrid modelling approach to generate large datasets of forest stands and corresponding hyperspectral data in the 400 nm-2400 nm range. Based on these datasets, we systematically evaluated the potential of vegetation indices to estimate above-ground biomass, LAI, GPP, and NPP. We found that estimates from data-driven indices could be significantly more accurate than predictions derived from "classical" vegetation indices.

We assessed which wavelength combinations were best suited for estimating the considered forest characteristics and observed that combinations of NIR and SWIR light yielded good results in general, with biomass, LAI, and GPP often being well estimable via the same wavelength combinations. The optimal choice of wavelengths depended on the structure of the considered forests, with visible light gaining in importance in less dense and structurally heterogeneous forests.

We proposed and evaluated a new class of vegetation indices, namely parameter-free vegetation indices. We found that the functional form of the vegetation indices did not significantly affect the maximal possible achievable accuracy, but instead constrained the range of wavelengths where this accuracy could be attained. We obtained a similar result with respect to uncertainty: evaluating different potential sources of uncertainty in physiological and environmental parameters, we observed that while uncertainty did not strongly reduce the achievable accuracy, it decreased the range of wavelengths where accurate vegetation indices could be constructed. These results, along with the simulation approach introduced in this study and the generated data, may facilitate the development of new data-driven vegetation indices, optimized for estimating individual forest characteristics of interest and tailored to the structure of the considered forests.

*Data availability.* The forest characteristics and reflectance profiles generated in this study can be found at the Zenodo public repository at https://zenodo.org/doi/10.5281/zenodo.16748241 (Fischer et al., 2025).

Author contributions. Samuel M. Fischer: Methodology, Software, Formal analysis, Investigation, Data Curation, Writing - Original Draft,
 Visualization. Rico Fischer: Conceptualization, Methodology, Writing - Review & Editing, Supervision, Project administration, Funding
 acquisition. Andreas Huth: Conceptualization, Methodology, Writing - Review & Editing, Supervision, Project administration.

Competing interests. The authors declare that they have no conflict of interest.

Acknowledgements. The authors gratefully acknowledge funding by the German Federal Ministry for Economic Affairs and Climate Action; Project 50EE 2235 – "SIMWALD" and helpful discussions with colleagues at the Department of Ecological Modelling at UFZ Leipzig.

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
