# Peer review of "Assessing forest properties with data-driven vegetation indices: insights from 900,000 forest stands"

_EGUsphere, 2025_

## Referee Comment (RC2)

Peer review report on "Assessing forest properties with data-driven vegetation indices: insights from 900,000 forest stands"

**Comments to Authors:**

**Overview and Major Comments:**

This manuscript presents a comprehensive and methodologically innovative study that systematically explores the potential of data-driven vegetation indices (VIs) to estimate key forest properties (biomass, LAI, GPP, and NPP). By coupling an individual-based forest dynamics model (FORMIND) with a multilayer radiative transfer model (mSCOPE) and applying a Monte Carlo sampling strategy, the authors generate an exceptionally large and well-structured synthetic dataset. The analysis of all possible two-wavelength combinations across the 400–2400 nm range, combined with an explicit treatment of multiple uncertainty sources, represents a substantial advance over existing studies.

Overall, the manuscript is scientifically sound, clearly written, and highly relevant to the remote sensing and forest ecology communities, particularly in the context of emerging hyperspectral satellite missions such as EnMAP. The study offers valuable conceptual insights into wavelength selection, index design, and uncertainty robustness. I think the manuscript suitable for publication after some revisions, mainly aimed at clarifying applicability and ensuring reproducibility.

**Major comments**

1. The introduction could be strengthened by improving accessibility for a broad audience. A greater emphasis on the ecological motivation would be beneficial. The authors could clarify why forest parameters such as forest biomass, LAI, GPP, and NPP are critical variables and why their large-scale estimation remains challenging.

2. While the comprehensive coupling of forest model and radiative transfer model is impressive, the manuscript would be more convincing if the authors could demonstrate that FORMIND reasonably represents real forest conditions. For example, how do the observation-based estimates of forest properties at sites such as "Hohes Holze" (i.e., biomass, LAI, GPP, NPP) compare with the simulated ranges generated by FORMIND? Are all observed values captured within the model's simulated distributions?

**Minor comments:**

Line 5: "two wavelengths (400 nm-2400 nm)"→"two wavelength (within 400 nm -2400 nm), to avoid the confusion.

Line 116: You may specify the meaning of the ODM and the correct unit for different variables.

Lines 117-118: Consider briefly explaining the interpretation of DBH entropy values—for example, does a more negative value indicate lower heterogeneity?

For Table 1: It would be helpful to add references or explanations regarding the rationale for selecting the wavelengths used in classical indices.

Lines 179-180: what are the criteria to select the "separating thresholds" for biomass and DBH entropy?

Lines 341-342: The study would be further strengthened if the developed hybrid model could be validated against sites with hyperspectral observations and observation-based estimates of forest properties in future work.

---

## Author Comment (AC1)

**Response to RC2**

This manuscript presents a comprehensive and methodologically innovative study that systematically explores the potential of data-driven vegetation indices (VIs) to estimate key forest properties (biomass, LAI, GPP, and NPP). By coupling an individual-based forest dynamics model (FORMIND) with a multilayer radiative transfer model (mSCOPE) and applying a Monte Carlo sampling strategy, the authors generate an exceptionally large and well-structured synthetic dataset. The analysis of all possible two-wavelength combinations across the 400–2400 nm range, combined with an explicit treatment of multiple uncertainty sources, represents a substantial advance over existing studies.

Overall, the manuscript is scientifically sound, clearly written, and highly relevant to the remote sensing and forest ecology communities, particularly in the context of emerging hyperspectral satellite missions such as EnMAP. The study offers valuable conceptual insights into wavelength selection, index design, and uncertainty robustness. I think the manuscript suitable for publication after some revisions, mainly aimed at clarifying applicability and ensuring reproducibility.

- Thank you for your time and effort in reviewing our manuscript and all your helpful comments! We respond to your individual comments below and make suggestions to improve the manuscript.

*Major comments*

1. The introduction could be strengthened by improving accessibility for a broad audience. A greater emphasis on the ecological motivation would be beneficial. The authors could clarify why forest parameters such as forest biomass, LAI, GPP, and NPP are critical variables and why their large-scale estimation remains challenging.

- Thank you. Your comment is in line with the other reviewer's comment, and we will address this in our revision. Specifically, we will rewrite the introduction's story line, putting a strong emphasis on the motivating underlying problem of estimating forest properties from remote sensing data. To that end, we will also touch at more of the existing literature in this area (e.g. Xiao et al., 2019). To make space for the additional text, we will reduce the background on methods and move it to the methods section where applicable.

2. While the comprehensive coupling of forest model and radiative transfer model is impressive, the manuscript would be more convincing if the authors could demonstrate that FORMIND reasonably represents real forest conditions. For example, how do the observation-based estimates of forest properties at sites such as "Hohes Holze" (i.e.,

biomass, LAI, GPP, NPP) compare with the simulated ranges generated by FORMIND? Are all observed values captured within the model's simulated distributions?

- Thank you for these questions. We have validated FORMIND at the "Hohes Holz" site by initializing it with forest inventory data and comparing the model predictions with values for GPP, NEP, and respiration derived from eddy covariance data at the site (Pohl et al., 2023). The validation results are presented in our supplement (S5.3). Further validation results on a daily time scale (but an earlier version of the parameterization) can be found at Holtmann et al. (2021). We will make these results more accessible by referring to them in the main text (line 97).

*Minor comments:*

Line 5: "two wavelengths (400 nm-2400 nm)"à"two wavelength (within 400 nm -2400 nm), to avoid the confusion.

- Will be fixed.

Line 116: You may specify the meaning of the ODM and the correct unit for different variables.

- We will do that.

Lines 117-118: Consider briefly explaining the interpretation of DBH entropy values—for example, does a more negative value indicate lower heterogeneity?

- We will do that. Yes, a lower value indicates lower heterogeneity and / or stronger dominance of a single individual.

For Table 1: It would be helpful to add references or explanations regarding the rationale for selecting the wavelengths used in classical indices.

- We chose wavelengths compatible with existing satellite missions (e.g. MODIS, Landsat), which typically cover larger bands rather than individual wavelengths. As such, the potential wavelengths are not unique. We will clarify in the text.

Lines 179-180: what are the criteria to select the "separating thresholds" for biomass and DBH entropy?

- We chose them so that the dataset would be split somewhat evenly (we chose rounded values for easier presentation). We will clarify this in the text

Lines 341-342: The study would be further strengthened if the developed hybrid model could be validated against sites with hyperspectral observations and observation-based estimates of forest properties in future work.

- We fully agree. Though outside the scope of this study, we will add more detailed recommendations for testing and developing new vegetation indices. This will

facilitate follow-up studies. Please see also our response to the corresponding point of reviewer 1.

**References**

The references can be found in the reviewed main text. The only new reference used in our response is the following:

Holtmann, A., Huth, A., Pohl, F., Rebmann, C., & Fischer, R. (2021). Carbon Sequestration in Mixed Deciduous Forests: The Influence of Tree Size and Species Composition Derived from Model Experiments. Forests, 12(6), 726. https://doi.org/10.3390/f12060726

---

## Author Comment (AC2)

**Response to RC1**

This manuscript examines the effectiveness of existing vegetation indices and novel hyperspectral indices for estimating forest properties: biomass, leaf area index (LAI), gross primary production (GPP) and net primary production (NPP). The methods combine individual-based forest and radiative transfer models to simulate many forest stands, their estimated biomass, LAI, GPP, NPP and 250 canopy wavelengths. Results suggest that unique wavelength pairs often offer stronger estimates of NPP, GPP and biomass than common vegetation indices. A data driven approach also offers strong correlations.

In general, I found the applied methodology interesting and compelling. The manuscript offers several valuable insights, for example, Table 2 could serve as a nice reference when considering existing vegetation indices for estimating forest properties. This said, in my opinion, the manuscript also offers significant room for improvement, particularly in its focus and clarity. I apologize in advance for my many comments below, but I hope they will aid the authors in their revisions. Overarching comments are followed by line-by-line comments.

- Dear Colin Bloom, thank you for your thorough and thoughtful review! We appreciate your comments and your efforts to help us improve our manuscript. Below we respond to your individual comments and make suggestions to adjust the manuscript accordingly. In the revision, we will put an emphasis on presenting the focus more clearly. In particular, we will shorten the introduction and focus more on the research gap and question. We will remove methodological explanations from the introduction.

Introduction: A systematic and data driven identification of important reflectance wavelengths for forest properties is absolutely warranted. Few conventional vegetation indices have been systematically developed to target important forest parameters like NPP or GPP. This is a compelling argument for conducting this analysis. Hyperspectral data and data driven indices derived from synthetic samples are only one way to investigate the problem. Much of the introduction, however, is taken up by a methodological background (Lines 33 to 74), which attempts to set the stage for methodological novelty. I find this argument less compelling. As mentioned, Henniger et al. (2023) has already applied the novel combined methodology used in the analysis. Tweaking the approach is just an application of the methods. In my opinion, the methodological background could be substantially condensed and placed at the start of the methods. In its place, a clearer introduction to the overarching problem and research question would greatly improve the clarity of the work. Some relevant questions which I think need to be addressed therein: What are NPP, GPP, and biomass and why are they important? Why do we need to monitor these parameters? How have

vegetation indices traditionally been developed? Have vegetation indices been produced for NPP or GPP specifically? What platforms exist (or will exist soon) to monitor changes? I believe more complete answers to these questions will make it abundantly clear why a new systematic approach to vegetation indices is needed.

- Thank you. Your comment is in line with the other reviewer's comment, and we will address this in our revision. We will rewrite the introduction's story line, putting a strong emphasis on the motivating underlying problem of estimating forest properties from remote sensing data. To that end, we will also refer to more of the existing literature in this area (e.g. Xiao et al, 2019). We will reduce the background on methods in the introduction and move it to the methods section where applicable.
- It is correct that the general ideas of (1) generating forest stands randomly using a Monte Carlo–type approach and (2) coupling a radiative transfer model with a dynamic forest model are inspired by earlier work. However, we would like to emphasize that our contribution goes beyond merely "tweaking" existing approaches. First, these two approaches had not previously been combined, which is why we believe the integration presented here is an original contribution that needs some introduction. Second, our forest stand generation technique employs a different algorithm than that of Henniger et al. (2023b) and exhibits different—and more rigorously specified—statistical properties. Although this aspect is not the main focus of the manuscript, we consider it important to mention briefly. We agree that this was not sufficiently clear in the introduction and will revise the manuscript accordingly.

Methods and Results: Understanding the manuscript in its entirety took several reads and more effort than I think is necessary. While I appreciate and understand the sensitivity testing of the radiative transfer model, this is, in my opinion, secondary to the overall intent of the main manuscript. Moving this portion of the analysis into the Supplement would make it much easier to read the manuscript through and to understand the methods and results.

- It is important to us to make the manuscript as accessible as possible, so thank you for this valuable feedback. Reading your comment, we believe that the motivation for this analysis might not have been laid out sufficiently clearly. Therefore, we would like to provide some further explanation for the analysis below, hoping that it will facilitate our discussion on how to find the best way forward in improving the paper.
- The typical purpose of a sensitivity analysis is to understand how strongly a model result changes if the input parameters change. The result is some range of confidence: even if we do not know the input parameter perfectly, the model results will remain valid to a specified degree. However, this is not what we do here. In our analysis, we provide advice on how **the model itself** (not its output) should change in the presence of uncertainty.

For example, under dry conditions, bare soil reflects the sunlight brightly, under wet conditions less so. Therefore, we may use a different vegetation index if we know the soil is dry than if we are uncertain. The same applies to the other analyzed factors: if we are certain of the chlorophyll or carotenoid content of leaves, we may use different wavelengths than if we are uncertain; if we know we are looking at a dense forest, we may use different wavelengths than when looking at a sparse forest or a forest whose density is variable.

The wavelength-$R^2$ map we show in Fig. 2 corresponds to the one case in which all parameters except the forest structure are certain. This is arguably a special case. Hence, to guide the appropriate selection of wavelengths for other applications, we would need to construct a similar figure for the respective expected uncertainty regime, which we do not know a priori. Therefore, we chose to provide a comprehensive analysis of many uncertainty scenarios and summarized each wavelength-$R^2$ map in one bar of Fig. 4 or one circle in Fig. 5, showing which areas of the wavelength-$R^2$ yield good vegetation indices.

- We hope that this explanation clarifies why we consider these results to be relevant. We will add a similar explanation to the revised paper, alongside corresponding adjustments to the discussion section.

Methods and Discussion: A lot of effort was made to evaluate different forest structures and sensitivity test the influence of different model parameters. However, it seems like the models are all based on the environmental conditions at one site in Germany (e.g., line 114). Does this also warrant a sensitivity test? At a minimum, it requires justification. Is this site representative of a broad range of sites? Is the model going to be overfit to the conditions at Hohes Holz? If not, why not?

- Thank you, this is a good point. An earlier analysis showed that the ForestFactory approach is able to cover the range of forests found in Germany (see Fig. 8 in Bohn and Huth, 2017), so that the generated dataset is expected to be representative for a broad range of forests. To limit the scope of the already complex manuscript, we decided not to randomize all environmental conditions, as they may interact in complex ways. A future study could build on our approach by randomly sampling environmental conditions for a given region of interest, but we do not expect that the additional randomization of the climate will lead to strongly deviating results, as we already randomized the environmental effects on the reflectance profile. Nonetheless, we will communicate the potential methodological extension regarding environmental conditions transparently, so that future work can build on this.

Methods and Discussion: It seems that all wavelengths are an average of the plot, but I cannot find this mentioned. This needs to be discussed. Higher and higher spatial resolution multi- and hyperspectral data are commonplace. Are the model results still

relevant on different scales? EnMAP, for example, has 30 m resolution. What is the influence of variability across the plot?

- We considered forest patches of 20m x 20m (see line 103), which is the natural scale of the FORMIND model and the same order of magnitude as the EnMAP data. An analysis across spatial scales would certainly be interesting and relevant, but we see particularly the EnMAP mission as our main motivating application. Hence, we believe it is fair to defer a scale analysis to future studies if more coarse-grain results are desired.

Methods and Discussion: Could a simple correlation analysis between parameters and single wavelengths provide some additional insight into appropriate wavelengths before looking at pairs?

- This is an interesting idea. Correlation analyses between parameters and single wavelengths have been carried out in several studies already, see e.g. Mousivand et al. (2014), Verrelst et al. (2015), and Prikaziuk and van der Tol (2019).
- If you are instead interested in an analysis of the correlation of single wavelengths with the considered forest properties, this could be done as well. We suggest refraining from adding more details to the already complex study, but if you considered it key, we would carry out the analysis and add it to the supplement.

Discussion: I believe this analysis would benefit immensely from (even a very small) application to real-world data. I realize that it is an additional effort which the authors may consider beyond the scope of the analysis, but it is relevant to the level of trust that can be placed in the primarily theoretical findings. Could several derived indices simply be applied to data from Hohes Holz to estimate biomass and compare with survey data? At a bare minimum, some concrete recommendations for next steps are required in the discussion. What models should we try with real-world data? For example, a short table of high performing wavelength combinations which match common satellite data could be very helpful. A huge amount of detailed effort has gone into the analysis, so it seems a bit of a shame for the discussion to end on high-level findings about general regions of the light spectrum that are good for GPP and NPP.

- Thank you for these suggestions. Of course, a direct application to ground data would be very desirable. This would require an exact mapping of EnMAP and ground-based data, appropriate pre-processing such as error correction of the ground data, and a thorough statistical analysis. We therefore would indeed consider this out of scope of this study. However, we will also clearly state this as a limitation of our study and thus motivate follow-up studies.
- In preparation of the manuscript, we did consider high-resolution hyperspectral data from the Hohes Holz site, which we downsampled to get a sense of the general model validity. However, the site is too small and too homogeneous for putting the derived vegetation indices to a meaningful test – the individual pixels are within the

general point cloud in the predicted-observed space, but too narrow together to derive a slope. We therefore decided not to include this Figure but rather to point to future studies for such an analysis.

- We really like the idea of providing more detailed recommendations for testing and developing new vegetation indices. Especially with our experiences mentioned above, we will be able to give helpful insights into how this process should work, which will then facilitate follow-up studies.

*Line-by-line Comments:*

Line 5: "with two wavelengths (400 nm- 2400 nm)" suggests only two wavelengths are evaluated. I think this should read something like: "using wavelengths between 400 nm and 2400 nm"

- Will be fixed.

Line 9: "provide highly accurate estimates" of what?

- Forest attributes. Will be fixed.

Line 9-11: This sentence is generally unclear and 'estimable' (which means 'worthy of great respect') was unlikely the intended word choice.

- We will adjust the wording to improve clarity and exchange "estimable" with "inferable".
- FYI, "estimable paramters" / "estimability" is an established concept in the stats literature.

Line 12: "did not primarily reduce the achievable accuracy" I am not sure what this means. The accuracy of what?

- The accuracy of the wavelength-based models for estimating forest properties. Will be fixed.

Line 21: "Here" implies that remote sensing is part of the analysis.

- We will remove the word.

Line 24: Just because hyperspectral data captures more data does not make it inherently more useful than traditional multi-spectral data. See my comment on the direction of the introduction above.

- Thank you. The sentence was intentionally formulated in the subjunctive mood, but the issue will be fixed with the revised introduction.

Line 55: 2500 nm does not match the rest of the text.

- Will be fixed.

Line 133: This seems somewhat circular but also makes a good case for applying new vegetation indices to the actual site to see how it performs.

- We agree. Please refer to our response to your last main comment.

Line 134 to 149: Moving this (and associated results) to the Supplement would help smooth out the main text.

- Thank you. Please refer to our response to your first comment on Methods and Results.

Line 151 to 152: How much was removed by this filtering?

- Less than 2%. Will be added to the text.

Line 161: sp. value

- Will be fixed.

Table 1: Different sensors capture different wavelengths, why were these chosen for classical indices? I think it deserves an explanation in the text.

- We chose wavelengths compatible with existing satellite missions (MODIS, Landsat) which typically cover larger bands rather than individual wavelengths. As such, the potential wavelengths are not unique. We will clarify in the text.

Line 182: "in the absence of noise" Is this the uncertainty analysis? It is not clear.

- Yes. We will clarify.

Line 195: Can this include a percent difference?

- Yes, we can provide quantitative results if desired.

Line 202: "estimable" again.

- See our comment above.

Line 280: Yes, but it was not the original intent of most VIs to determine NPP or GPP. This should be clarified here and in the introduction.

- This is true, and we will do this.

Thank you for the opportunity to evaluate this work and I wish the authors the best of luck in their revisions.

- Thank you, we are very happy to have you as reviewer and will do our best to improve the manuscript.

Colin Bloom

**References**

The references can be found in the reviewed main text.